# Mix-VIO: A Visual Inertial Odometry Based on a Hybrid Tracking Strategy

**DOI:** 10.3390/s24165218

**Published:** 2024-08-12

**Authors:** Huayu Yuan, Ke Han, Boyang Lou

**Affiliations:** School of Electronic Engineering, Beijing University of Posts and Telecommunications, Beijing 100876, China; yuanhuayu@bupt.edu.cn (H.Y.); hanke@bupt.edu.cn (K.H.)

**Keywords:** visual-inertial odometry, deep learning-based feature detection, hybrid tracking strategy

## Abstract

In this paper, we proposed Mix-VIO, a monocular and binocular visual-inertial odometry, to address the issue where conventional visual front-end tracking often fails under dynamic lighting and image blur conditions. Mix-VIO adopts a hybrid tracking approach, combining traditional handcrafted tracking techniques with Deep Neural Network (DNN)-based feature extraction and matching pipelines. The system employs deep learning methods for rapid feature point detection, while integrating traditional optical flow methods and deep learning-based sparse feature matching methods to enhance front-end tracking performance under rapid camera motion and environmental illumination changes. In the back-end, we utilize sliding window and bundle adjustment (BA) techniques for local map optimization and pose estimation. We conduct extensive experimental validations of the hybrid feature extraction and matching methods, demonstrating the system’s capability to maintain optimal tracking results under illumination changes and image blur.

## 1. Introduction

In recent years, Simultaneous Localization and Mapping (SLAM) systems have been widely applied in robotics, drones, autonomous driving and AR/VR [1,2]. Visual SLAM, benefiting from its small sensor size and low power consumption, has received extensive and long-term research [3]. As a branch of visual SLAM, visual-inertial odometry (VIO), can compensate for image blur to some extent by integrating information from IMU sensors to achieve higher accuracy than a conventional visual SLAM system. Over the past decade, numerous outstanding VIO systems have been introduced, such as the ORB-SLAM series [4,5,6,7], DSO series [8,9] and VINS [10], etc. The front end of these systems often uses feature-based methods or direct methods to track the landmarks and acquire point correspondences between frames, thereby obtaining sufficient inter-frame constraints to optimize the state estimation problem. So, it stands that good front-end tracking performance is crucial to the hall visual odometry system. After more than 10 years of development, the conventional front-end tracking methods can be summarized into three types: traditional handcraft approaches, deep learning approaches and hybrid approaches.

The traditional schemes of visual-inertial odometry systems are mainly divided into direct methods and feature-based methods. Direct-method approaches assume constant inter-frame grayscale values to derive the transformation relationships between frames. As early as 30 years ago, the LK optical flow [11] tracking approach was proposed and widely used in image stitching and feature tracking. Based on the assumption of grayscale invariance, L. and K. and others posited that the pixel grayscales remain constant and unchanging between two instants, thereby deriving sparse or dense optical flow equations to obtain inter-frame matching results. In recent years, this method has been extended to direct pose optimization approaches, such as [9,10], achieving impressive results. The direct method holds under conditions of high frame rates and stable environments because the grayscale changes between frames can be approximated as nonexistent, the grayscale assumption usually holds and, considering an entire block of pixels, the system has enhanced robustness against image blur. However, under complex lighting conditions, this approach often fails, frequently leading to loss of system tracking, as shown in Figure 1.

To address these issues, researchers turned to another intuitively appealing branch: the feature-based method. By identifying and describing prominent keypoints or other representative features in images, the system can extract more useful information from images and use this information to match similar features between frames. The earliest keypoint detection and matching algorithms can be traced back to the PTAM-SLAM [14], which is also considered the first comprehensive feature-based SLAM system. Subsequently, based on ORB [5] keypoint detection methods, ORBSLAM was proposed and then followed by the development of ORBSLAM2 and ORBSLAM3. With FAST [15] keypoint detection and robust rotation-invariant BRIEF description methods, the ORBSLAM series are very robust and stable in front-end feature tracking, marking another milestone in the SLAM systems.

Although the tracking and the localization performance are excellent, systems based on feature points slightly underperform in tracking results when encountering image blur. In ORBSLAM3 [8], the authors themselves mentioned that inter-frame matching results using optical flow tracking are superior to those using manual designed keypoint extraction and matching methods; on one hand, because the parameter and patterns are artificially designed, even when using feature enhancement techniques like image pyramids, they are fixed according to certain templates, hence the detection result cannot automatically adapt to the entire image, which is already slightly less effective than optical flow tracking methods. On the other hand, under high-speed camera motion, image blur is inevitable, resulting in poor performance of the descriptor-based keypoint matching methods under such conditions, while the direct method based on image block and iterative tracking performs better. Therefore, we combine both the direct method and feature extraction approaches, which is not uncommon in traditional schemes as seen in [16], which uses semi-direct methods for system tracking, or such as [17], which used direct methods to accelerate feature matching. An alternative approach is to first extract keypoints and then use sparse optical flow for feature-tracking matching; many VIO systems adopt such schemes, like [10,18,19], all achieving excellent localization results.

In our system, we refer to the aforementioned VIO systems [10], and combine optical flow tracking and feature point matching as our front-end matching scheme. Different to [16,17], we combined the optical flow-tracking direct method and deep learning-based feature extraction and matching pipeline in parallel to generate the final inter-frame correlation relationship.

With the advent of deep learning technology, numerous SLAM schemes based on deep learning have been proposed to address challenges in varying lighting and dynamic environments. Initially, researchers often modeled the entire bundle adjustment (BA) optimization problem or pose estimation problem as a complete network [20,21,22], supervising the entire pipeline with true pose values. Although this approach achieved a fairly decent localization accuracy on synthetic or real datasets, it often faced many issues when transplanted into real-world scenes, with the tracking accuracy also not being guaranteed. With the release of many studies and papers, it has generally been accepted that these fully network-based end-to-end pose prediction methods are impractical, leading researchers to shift their focus back onto traditional SLAM pipelines [23]. Researchers began considering integrating some modules of traditional SLAM with deep learning solutions, such as feature extraction and matching methods [24] and integrating into the tracking front-end.

As such, the deep learning-based feature point detection and matching have become hot topics. The earliest deep learning-based feature point detection network can be traced back to TILDE [25], which trained a model using a camera with a fixed perspective to collect data on different times and weather conditions for training, and then utilized CNN to solve the problem where traditional feature detection operators are always sensitive to weather and lighting changes. It enhanced the repeatability of keypoints, paving the way for subsequent algorithms. Following TILDE, numerous deep learning-based feature point detection networks like Lift [26] and feature descriptor networks [27] were proposed, often based on complex network structures such as Siamese networks or intricate design processes, which resulted in long inference times. Moreover, most feature point or keypoint detection networks still relied on traditional SIFT [28] features for network supervision, making them impractical in real applications. It was not until 2018 that Superpoint [29] was introduced, which proposed a self-supervised, fully convolutional model for keypoint detection and description. It utilized homography adaptation to improve the accuracy and repeatability of feature point detection and achieved cross-domain adaptation from simulated datasets to the real world. Due to its simple structure, lightweight network and strong robustness to lighting, it has been widely used in the field of SLAM. For instance, Superpoint-based SLAM [30], which replaced the manual features in the ORB-SLAM system with Superpoint features, achieved better localization results.

In subsequent research, the author of Superpoint introduced a network based on graph transformers named Superglue [31], which significantly enhanced the overall effect of feature extraction and matching. As an improved version of Superglue, Lightglue [32], utilizing stacked transformer layers and reasoning techniques for early exit determination, achieved superior matching performance and faster speeds in image-matching tasks. However, when transitioning to SLAM, directly using deep learning-based feature extraction and matching networks into the scheme proved challenging due to the high inference cost, often unable to achieve real-time front-end performance. Our solution integrates the Superpoint and Lightglue networks to enhance our front-end feature-tracking performance, using the TensorRT neural network library to accelerate the network inference to ensure the real time performance of the network.With the rise of deep learning for feature extraction and matching, many researchers are considering the integration of deep learning methods with traditional approaches to achieve better feature-tracking results. A notable example is the D2Vins proposed by Xu, H from HKUST, which is used as the state estimation scheme in the D2SLAM [33] swarm state estimation system. It utilizes the Superpoint network in the front end to produce a feature map with depth descriptor feature information, and then traditional manually designed feature extractors like Good-Feature-to-Track (GFT [34]) are used to detect the keypoints. Subsequently, the positions of the keypoints are used to sample the entire feature map and obtain the corresponding descriptors for each point. These descriptors are then used for loop closure detection in a single UAV and feature association between different UAVs with a mutual visibility relationship, enabling the robust feature correlation and state estimation between the drone swarm. In Airvo [35], the author implemented both deep learning-based Superpoint feature point extraction and LSD [36] line segment detector, using Superglue to match Superpoint point features and utilizing Superpoint for line feature encoding to enhance line feature descriptions through Superpoint’s illumination invariance, thus achieving robust lighting matching for line features.

In recent years, in addition to the three tracking schemes mentioned above, there has been a notable surge in the development of VIO systems that rely on multiple visual sensors or innovative types of visual sensors. Multi-camera systems [37,38], for example, use an array of cameras to achieve a larger field of view and more stringent feature constraints, resulting in more accurate tracking and localization performance. In conditions of drastic lighting changes, other cameras in the multi-camera system can compensate for the exposure failures of individual cameras, thereby enhancing the system’s robustness. The event camera-based visual-inertial odometry system [39,40], which leverages biomimetic vision, operates differently from traditional methods. Event cameras detect changes in brightness at the pixel level independently and transmit signals only when these changes occur, generating asynchronous event streams with microsecond-level precision. Each event is characterized by its time-space coordinates and occurrence flag, rather than intensity. Because event cameras capture changes in brightness over time, they excel at detecting edge transitions in rapidly moving scenes, offering superior performance in challenging conditions such as dynamic lighting and motion blur. However, these systems are currently quite expensive and have not been widely adopted, with their popularity lagging behind that of conventional monocular or binocular cameras. Additionally, their system construction is more complex.

In contrast to the aforementioned solutions, our approach centers on conventional monocular or binocular imaging systems. Unlike Airvo [35], which combines deep learning-based keypoint detection and the traditional line detection methods, we focus exclusively on point features and employ an enhanced version of Superglue, called Lightglue, for feature matching. We also utilize traditional manually designed keypoints for inter-frame optical flow tracking, which operates independently from Superpoint + Lightglue deep feature extraction and matching pipelines, without encoding traditional points with deep points. In contrast to D2Vins [33], which relies on deep learning for feature extraction and matching, we directly extract keypoints using the Superpoint newtwork and apply the Lightglue feature matcher for inter-frame feature matching. These inter-frame relations are incorporated into the overall state estimation problem to reinforce constraints. For traditional features with strong gradients, we perform optical flow tracking to achieve superior results, rather than solely relying on Superpoint as a descriptor network. Our experiments revealed that traditional optical flow tracking remains effective in scenarios with blurry images, whereas deep learning methods tend to fail more frequently (as shown in Figure 1). This underscores the importance of including manually designed features in the system. Additionally, we evaluated the reliability of depth features versus manual matching methods and designed a feature point dispersal strategy that integrates both deep and manual features.

To address issues caused by varying lighting conditions and image blur, this paper proposes a visual-inertial odometry (VIO) system that integrates deep feature extraction and matching with traditional optical flow matching. Extensive and robust comparative experiments were conducted within the same framework to evaluate both the traditional optical flow method and the deep learning-based feature extraction and matching pipeline. After thorough analysis, we implemented a fusion approach of the two methods. The experiments demonstrate the advanced nature and robustness of our system.

In summary, our contributions are as follows:We propose a VIO system that is robust for illumination changes and accurate in tracking. To tackle dynamic lighting and high-speed motion environments, we combine deep learning with traditional optical flow for feature extraction and matching, presenting a hybrid feature point dispersion strategy for more robust and accurate results. Leveraging TensorRT for parallel acceleration of feature extraction and matching networks enables real-time operation of the entire system on edge devices.Unlike the aforementioned approaches [16,17] that accelerate optical flow tracking using direct methods, our approach combines optical flow with parallel depth feature extraction and feature matching. We employ a hybrid method of optical flow tracking and feature point matching as our front-end matching scheme, achieving robustness against image blurring and lighting changes.We have open-sourced our code at https://github.com/luobodan/Mix-vio (accessed on 12 June 2024) for community enhancement and development.

## 2. Materials and Methods

### 2.1. System Overview

As shown in Figure 2, our system architecture is based on VINS-fusion. To achieve feature robustness under dynamic illumination, we utilize the Superpoint feature extractor and utilize the most advanced DNN-based sparse feature matcher Lightglue to match features between frames. In our system, the network parameters are not fine-tuned, relying entirely on pre-trained models for experimentation. Considering the blurring problems of the camera under high-speed motion, traditional GFT [34] features and bi-directional tracking-assisted features are used for inter-frame tracking. A feature-dispersal approach that considers both optical flow tracking results and Lightglue feature matching results is used. After successful feature tracking, these successfully tracked features are sent to the backend optimizer for optimization processing. The Visual-IMU initialization strategy is consistent with [10]. When an IMU frame arrives, the frontend calculates state propagation and performs prediction processing, passing the IMU prediction components to the backend for BA joint optimization, using a sliding window to control the scale of the overall optimization problem.

### 2.2. DNN-Based Feature Extraction and Matching Pipeline Based on Superpoint and Lightglue

The deep feature extraction and matching pipeline is very fast and simple. We use Superpoint as the feature extractor and use Lightglue, the state-of-the-art lightweight sparse feature matching net, for deep feature matching. For faster inference speed, we accelerate the network based on TensorRT(Company Name: NVIDIA Corporation Address: 2788 San Tomas Expressway, Santa Clara, CA 95051, USA), as shown in Figure 3.

#### 2.2.1. Deep Feature Detection and Description Based on Superpoint

As shown in Figure 4, the Superpoint network architecture employs a VGG network for feature extraction to generate feature maps, and uses an encoder–decoder module to process these maps, outputting final coordinates and descriptors for feature points. The network takes an input image ‘image’ with size H×W×C, initially processes it through grayscale thresholding, and normalizes the grayscale values globally to produce a grayscale normalized image I with one channel and resolution H×W×1. The shared encoder layer, comprising consecutive convolutional kernels, activation functions and max pooling layers, expands channels through convolution of the extracted VGG features, and downsamples the resolution through pooling to a tensor feature of size (H/8×W/8×128).

For the feature point extract branch, the encoder layer’s output feature is appended to a CNN-based decoder, and finally outputs a feature tensor sized (H/8×W/8×65), where 65 = 64 + 1 indicates the possibility of each position being a feature point within an 8 × 8 area, and the extra channel represents a “garbage can”, indicating the probability of no features in that area. Following this, the tensor is passed through a softmax and reshape function to obtain a final feature point probability map p with size H×W. The final set of n feature points and their scores are obtained by threshold filtering and non-maximum suppression based on this probability map p.

For the descriptor extraction branch, the encoder layer’s output feature is also appended to a CNN-based decoder, and produces a descriptor tensor D with resolution (H/8×W/8×256). The positions of the n feature points determined earlier from another branch are used to compute their corresponding 256-dimensional descriptors through bi-cubic interpolation.

The overall feature point extraction and descriptor network can be abstracted into function $f_{point}$ and $f_{des}$. For the input normalized image, there is
(1)$$f_{point}\left(I\right)=p_{I:n\times 2},\;{score}_{I:n\times 1}$$
(2)$$f_{des}(I,p_{I:n\times 2})={des}_{I:n\times 256}$$
where $p_{I}$ and ${des}_{I}$ represent the feature points and the descriptor, and ${score}_{I}$ represents the points scores.

#### 2.2.2. Deep Feature Matching Based on LightGlue

The structure of the LightGlue network consists of m identical layers, each based on Transformer architecture, that cascade to process a pair of images and corresponding pairs of feature points and descriptors, as shown in Figure 5. Each layer is composed of self-attention units and cross-attention units that update the representation of each point. A classifier at the end of each layer decides whether to halt the overall inference of the system, thereby saving computational time and ultimately outputting a similarity matrix at the stop position.

Specifically, once the feature points and descriptors $\left\{p_{I_{a}},{des}_{I_{a}}\right\},\{p_{I_{b}},{des}_{I_{b}}\}$, whose indices are A:={1,…,M} and B:={1,…,N}, respectively, for the original images have been input into the network, the descriptors’ state will be initialized to a state variable ${Χ}_{I}^{i}$. The ${Χ}_{I}^{i}$ encodes the feature information and is fed into subsequent transformer structure for iterative updates in each layer. This initialization process can be represented as
(3)$${Χ}_{I}^{i}\leftarrow {des}_{I}^{i}$$
where i iterates each feature, and I∈{A,B} denotes I belonging to the image A or image B and the S∈{A,B} corresponding to the source image, which is same to I during the self-attention module. The associated feature quantity ${Χ}_{I}^{i}$ between the two images is first updated by each independent transformer layer at the beginning of an MLP network:(4)$${Χ}_{I}^{i}\leftarrow {Χ}_{I}^{i}+MLP(\;[{Χ}_{I}^{i}|m_{i}^{I\leftarrow S}])$$
where [·] denotes the aggregation vector. The message is calculated by using the following equation:(5)$$m_{I\leftarrow S}^{i}=\sum_{j\in S}Softmax({a_{IS}^{ik}}^{j}{WΧ}_{S}^{j})$$
where W is the projection matrix and $a_{IS}^{ik}$ is the attention score of the points i and points j regarding the image I and S, calculated specifically according to the mechanisms of self-attention and relevant attention.

Based on the mechanisms of the transformer in self-attention units, each point within the same image attentively observes every other point in that image through self-attention. In this process, the representations of ${Χ}_{I}^{i}$ or ${Χ}_{S}^{i}$ are first linearly transformed to project into key and query vectors $k^{i}$ and $q^{i}$, respectively, and they utilize rotational encoding R(·) for encoding the relative positions between points $p_{j}$ and $p_{i}$. It is noteworthy that rotational encoding emphasizes more on the relative positioning of points rather than an absolute positional encoding, defining self-attention scores $a_{ij}$ as follows for features i and j:(6)$$a^{ij}={q^{i}}^{T}R\left(p^{j}-p^{i}\right)k^{j}$$
(7)$$R(p)=\left(\begin{array}{cccc} \hat{R}(b_{1}^{T}p) & & 0 & \\ & \ddots & & \\ 0 & & \hat{R}(b_{d/2}^{T}P) & \end{array}\right)$$
(8)$$\hat{R}(\theta )=\left(\begin{array}{cc} cos\theta & -sin\theta \\ sin\theta & cos\theta \end{array}\right)$$
where $b_{n}^{T}$ is the transpose coefficient, which is cached during all network inference calculations.

In the cross-attention units, each point in image I leverages the connection with each point in the other image, only calculating attention scores concerning the value vectors:(9)$$a_{IS}^{ij}=k_{I}^{i⊤}k_{S}^{j}$$

On correspondence prediction, a lightweight head prediction task assignment is designed. Firstly, the network calculates the match scores matrix $S\in R^{M\times N}$ between the two images as
(10)$$S_{ij}=\mathrm{Linear}{\left({Χ}_{A}^{i}\right)}^{⊤}\mathrm{Linear}\left({Χ}_{B}^{i}\right)\;\forall (i,j)\in A\times B$$
where Linear is a learned linear transformation with bias, computing the relevance scores between two 2D points. These scores are normalized to obtain the matching scores between feature points:(11)$$\sigma_{i}=\mathrm{Sigmoid}\left(\mathrm{Linear}\left({Χ}^{i}\right)\right)\in [0,1]$$

Thus, the entire LightGlue network can be abstracted as a function L, given the network inputs of extracted feature points and descriptors from both images, $p_{I_{a}:n\times 2}$, ${des}_{I_{a}:n\times 256}$, $p_{I_{b}:n\times 2}$, ${des}_{I_{b}:n\times 256}$, the network outputs a pair of matching feature points (these point pairs can be mapped back sequentially to the original input feature points, thus forming a one-to-one correspondence) along with their corresponding match scores:(12)$$L\left(p_{I_{a}:n\times 2},\;{des}_{I_{a}:n\times 256},\;p_{I_{b}:n\times 2},\;{des}_{I_{b}:n\times 256}\right)=pair(p_{1},p_{2},match\_score)$$
where $p_{1}$ and $p_{2}$ are the corresponding 2d points pair, match_score means the match scores.

### 2.3. Hybrid Feature-Tracking Strategy

Traditional optical flow tracking schemes employ GFT feature and image pyramids to track optical flow features between frames, as shown in the blue area in Figure 6. With pyramid feature scaling and the assumption of illumination invariance, sub-pixel-level feature matching results can be achieved. This is widely recognized as one reason why they are more accurate than feature point methods that merely extract feature points and descriptors for tracking. Another reason is that they register information from around the feature points, introducing iterative methods to iteratively track the positions of these points, thus making the optical flow method highly robust against blurry environments. However, traditional optical flow tracking schemes also have problems, namely, when there is significant illumination change, the assumption of illumination invariance often fails, leading to tracking divergence and making the system extremely non-robust.

The deep feature extraction and matching pipeline, trained on a large amount of image data with varying lighting conditions and different perspectives, possesses stronger robustness to lighting variations compared to traditional methods to some extent. However, they also have certain issues, such as the blurriness in images under rapid movement, as shown in the Figure 1. This can result in poor backend optimization effects (as seen in our experiment section). Thus, we have combined optical flow tracking with deep feature extraction to enhance feature extraction and matching. Our approach is inspired by the idea of D2SLAM [33], but unlike D2Vins, we use a hybrid of SP + LG and optical flow tracking schemes for direct feature matching and tracking, rather than solely relying on optical flow tracking and using SP as a feature descriptor. This is thanks in part to our use of the high-speed tensorRT library to accelerate the network, achieving a processing speed of 7 ms/frame. As shown in Figure 6, our strategy mainly contains two branches, the optical flow branch and the deep detection and feature matching branch.

On the optical-flow branch, when the camera collects a new frame image at $t_{k}$, the system firstly uses the optical flow to track the 2D coordinates of the optical flow tracking point $p_{I_{k-1}}^{opt}$ which cached in the upper moment $t_{k-1}$. In order to make the overall processing speed faster, we use the reverse compose optical flow method based on the multi-layer image pyramid to improve the overall operation efficiency. Inverse compositional optical flow [11] can be defined as
(13)$$\underset{p}{min}\sum_{x}{\left[I_{k-1}(W(p_{I_{k-1}}^{opt};∆q))-I_{k}(W(p_{I_{k-1}}^{opt};q))\right]}^{2}\;W(p_{I_{k-1}}^{opt};q)=\left(\begin{array}{c} u_{I_{k-1}}^{opt}+u\\ v_{I_{k-1}}^{opt}+v \end{array}\right)$$
(14)$$W(p_{I_{k-1}}^{opt};q)\leftarrow W(p_{I_{k-1}}^{opt};∆q)\circ {W(p_{I_{k-1}}^{opt};∆q)}^{-1}$$

To solve the above problem, the Formula (13) is expanded via the Taylor expansion in the point $W(p_{I_{k-1}}^{opt};0)$ and is solved iteratively using the Gaussian Newton method:(15)$$\underset{p}{\min}\sum_{x}{\left[I_{k-1}(W(p_{I_{k-1}}^{opt};0))+\nabla I_{k-1}(W)\frac{𝜕W}{𝜕q}∆q-I_{k}(W(p_{I_{k-1}}^{opt};q))\right]}^{2}$$
(16)$$∆q=H^{-1}\sum_{x}{{\left[\nabla I_{k-1}\frac{𝜕W}{𝜕q}\right]}^{T}\left[I_{k-1}(W(p_{I_{k-1}}^{opt};q)-I_{k}(p_{I_{k-1}}^{opt})\right]}^{2}$$
(17)$$H=\sum_{x}{\left[\nabla I_{k-1}\frac{𝜕W}{𝜕q}\right]}^{T}\left[\nabla I_{k-1}\frac{𝜕W}{𝜕q}\right]$$

Here, W is the image block warp map, which can be simply defined as the pixel addition and subtraction. q is the amount of pixels in the region, and ∆q is the pixel amount of the medium map transformation updated for the optical flow tracking. The pixel value I is corresponding to the image block near the feature point. Due to the expansion around $W(p_{I_{k-1}}^{opt};0)$, the matrix $\nabla I_{k-1}\left(W\right)$, $\frac{𝜕W}{𝜕q}$ and H can be calculated and cached at the beginning, thus saving more computing resources. The overall optical flow process uses the image pyramid to converge step by step. At the same time, for a pair of feature points on the optical flow tracking, we conduct bidirectional optical flow tracking and suppress the number of features near the points where the bidirectional tracking is successful to realize the decentralized processing of features.

On the deep detection and feature matching branch, we apply the SP feature extraction network on the image $I_{k}$ to extract n feature points $p_{I_{k}:n\times 2}$ and took out the feature points $p_{I_{k}:m\times 2}$ with the highest m score (the typical value of m is 1024, if the number of feature points is less than m and takes all the feature points into consider), and performed feature description to obtain descriptors:(18)$$f_{point}\left(I_{k}\right)=p_{I_{k}:n\times 2},{\;score}_{I_{k}:n\times 1}$$
(19)$$f_{des}\left(I_{k},p_{I_{k}:m\times 2}\;\right)={des}_{I_{k}:m\times 256}$$

Then, the feature points $\{p_{I_{k}:m\times 2},$ ${des}_{I_{k}:m\times 256}\}$ extracted from the current frame and the feature points $\left\{p_{I_{k-1}:m\times 2},{des}_{I_{k-1}:m\times 256}\right\}$ corresponding to the previous frame are sent together into the Lightglue network for matching:(20)$$L\left(p_{I_{k}},{des}_{I_{k}},p_{I_{k-1}},{des}_{I_{k-1}}\right)=pair(p_{1},p_{2},match\_score)$$

The network will output the feature pairs $pair(p_{1},p_{2},match\_score)$ matching the feature points between two frames, and then filter the point pairs with a matching score less than the th (typical value of 0.7):(21)$$if\left\{\begin{array}{c} match_{score}<th,\\ else, \end{array}\right.\;\begin{array}{c} delete\\ reserve \end{array}$$

When the number of matching feature points is fewer than the threshold of points, Superpoint features will be added to the back end for initialization and used as the basis for the next frame match. Once the SP + LG matching is completed, we use the geometry consistency to trim the matching point set. Subsequently, we sort the feature points based on the count of successful tracking and process them using radius feature features [10]. It is worth noting that, although these points are not added to the back end after dispersion optimization processing, they will still be appended to the Lightglue network for matching in the next deep feature matching pipeline. It will improve the system performance if the optical flow tracking fails or the suppression radius leaves the tracking point, which can significantly increase the feature to track.

If the bidirectional optical flow tracking succeeds, the feature point tracking matching will be more robust and more accurate than the only deep feature point matching, and we would take more optical-flow points into consideration. In our feature point dispersion strategy, we prioritize optical flow tracking success within a radius near the circle. Then we suppress optical flow on Superpoint features and aim to maximize the distance for deep features and bidirectional optical flow tracking feature points.According to the deep features, they are ranked according to the number of successful matches between frames and used to draw circles to suppress the deep points successively. The features in the circle are eliminated and not added to the back-end optimization. It is worth noting that because of this convenient end-to-end feature matching structure, we quickly obtain the matching relationship between feature points directly at once, which is an advantage of the traditional methods.

During optical flow tracking, the feature points with the highest score in each region were selected for optical flow tracking, and the feature points were dispersed according to the radius, often larger than it.

The pseudo-code for feature dispersion Algorithm 1 can be written as follows:
**Algorithm 1** feature points dispersion
**Input**: The successful optical flow tracking point vector p1 and the successful deep matching SP point vector p2, where p1 and p2 are sorted by the tracked times. And the min distance between the points, ropt for optical flow points, rsp for SP points.**Output**: the point set to add to the optimization pall
1Cv::Mat mask1, mask2; mask1.fillin(255); mask2.fillin(255); Vector pall;//Step 1. Construct the initial mask for p1 and p2. Construct pall
2for *p* in p1:3 if(mask1.at(*p*) == 255):4  Circle(mask1, −1, p, ropt);5  Circle(mask2, −1, p, rsp);6  p.use = true;7  pall. push_back(p);8 else: 9  p.use = false;10 end if11end for12for *p* in p2:13 if(mask2.at(*p*) == 255):14  p.use = True;15  Circle(mask2, −1, p, rsp);16  pall. push_back(p);17 else:18   p.use = false;19 end if20end for21return pall


At the same time, in order to make full use of the CPU resources on embedded devices, our system can also conduct multi-thread parallel acceleration of optical flow matching and feature matching network, so that the total time consumption of feature network matching and optical flow tracking decreases. Flag use indicates whether the backend optimization should include each point. The feature point dispersion process starts from the result of optical flow tracing, first drawing the circle of the optical flow tracing result, and assigning the corresponding area of mask1 and mask2 with different radius $r_{opt}$ and $r_{sp}$ to −1. Then, mask2 decentralizes the remaining feature points to form more scattered feature points. When the system contains binocular input, in order to save resources, we only perform the SP + LG+ pipeline on the left camera, and use the optical flow to track from the left to the right, while the feature point dispersion strategy is consistent with that described above.

#### 2.3.1. IMU State Estimation and Error Propagation

The kinematic equation for the IMU can be written as
(22)$$\dot{R}=R\omega \;\dot{p}=v\;\dot{v}=a$$

Among them, R and p means 3 × 3 rotation matrix and the position vector, during which ω, a and v mean the 3 × 1 angular velocity vector, 3 × 1 acceleration vector and 3 × 1 speed vector. However, considering the gravity, the bias and noise of the acceleration and angular accelerometer in IMU, the real acceleration and angular velocity measurement can be written as
(23)$${\hat{a}}_{t}=a_{t}+b_{a_{t}}+R_{t}^{w}g^{w}+n_{a}\;{\hat{\omega }}_{t}=\omega_{t}+b_{\omega_{t}}+n_{\omega }$$
where $\cdot_{t}$ means the state of the parameter at the moment in time t, $\cdot^{w}$ means the parameter is in world coordinate system. In (24), the noise $n_{a}\sim N(0,\sigma_{a}^{2})$ and noise $n_{\omega }\sim N(0,\sigma_{\omega }^{2})$. Additionally, 3 × 1 vectors $b_{a_{t}}$ and $b_{\omega_{t}}$ are modeled as the random walk process:(24)$$\dot{b_{a_{t}}}=n_{b_{a}}\;\dot{b_{\omega_{t}}}=n_{b_{\omega }}$$

Following the literature [12,13] using IMU pre-integral and ESKF to infer the error states, using Euler integral,
(25)$$\alpha_{b_{k+1}}^{b_{k}}=\iint_{t\in \left[t_{k},t_{k+1}\right]}R_{t}^{b_{k}}\left({\hat{a}}_{t}-b_{a_{t}}\right)dt^{2}$$
(26)$$\beta_{b_{k+1}}^{b_{k}}=\int_{t\in \left[t_{k},t_{k+1}\right]}R_{t}^{b_{k}}\left({\hat{a}}_{t}-b_{a_{t}}\right)dt$$
(27)$$\gamma_{b_{k+1}}^{b_{k}}=\int_{t\in \left[t_{k},t_{k+1}\right]}\frac{1}{2}\Omega \left({\hat{\omega }}_{t}-b_{\omega_{t}}\right)\gamma_{t}^{b_{k}}dt$$
where
(28)$$\Omega (\omega )=\left[\begin{array}{cc} -\lfloor \omega \rfloor_{\times } & \omega \\ -\omega^{T} & 0 \end{array}\right],\lfloor \omega \rfloor_{\times }=\left[\begin{array}{ccc} 0 & -\omega_{z} & \omega_{y}\\ \omega_{z} & 0 & -\omega_{x}\\ -\omega_{y} & \omega_{x} & 0 \end{array}\right]$$

The covariance matrix $P_{b_{k+1}}^{b_{k}}$ of 3 × 1 vectors α, β and quaternion-vector γ also propagates accordingly. It can be seen that the preintegration terms can be obtained solely with IMU measurements by taking $b_{k}$ as the reference frame given bias.

Based on the assumption of the constant bias during the two time moment between the time k and the k+1, we adjust $\alpha_{b_{k+1}}^{b_{k}},\beta_{b_{k+1}}^{b_{k}}$ and $\gamma_{b_{k+1}}^{b_{k}}$ by their first-order approximations with respect to the bias as
(29)$$\alpha_{b_{k+1}}^{b_{k}}\approx {\hat{\alpha }}_{b_{k+1}}^{b_{k}}+J_{b_{a}}^{\alpha }\delta b_{a_{k}}+J_{b_{w}}^{\alpha }\delta b_{w_{k}}$$
(30)$$\beta_{b_{k+1}}^{b_{k}}\approx {\hat{\beta }}_{b_{k+1}}^{b_{k}}+J_{b_{a}}^{\beta }\delta b_{a_{k}}+J_{b_{w}}^{\beta }\delta b_{w_{k}}$$
(31)$$\gamma_{b_{k+1}}^{b_{k}}\approx {\hat{\gamma }}_{b_{k+1}}^{b_{k}}⊗\left[\begin{array}{c} 1\\ \frac{1}{2}J_{b_{w}}^{\gamma }\delta b_{w_{k}} \end{array}\right]$$

At the same time, the pose of the system can be compensated by high frequency IMU:(32)$$P_{b_{k+1}}^{b_{k}}=\iint_{t\in \left[t_{k},t_{k+1}\right]}R_{t}^{b_{k}}\left({\hat{a}}_{t}-b_{a_{t}}\right)dt^{2}$$
(33)$$\theta_{b_{k+1}}^{b_{k}}=\int_{t\in \left[t_{k},t_{k+1}\right]}\frac{1}{2}\Omega \left({\hat{\omega }}_{t}-b_{w_{t}}\right)\gamma_{t}^{b_{k}}dt$$
while considering the error transfer between the systems:(34)$$\left[\begin{array}{c} \delta {\dot{\alpha }}_{t}^{b_{k}}\\ \delta {\dot{\beta }}_{t}^{b_{k}}\\ \delta {\dot{\theta }}_{t}^{b_{k}}\\ \delta {\dot{b}}_{a_{t}}\\ \delta {\dot{b}}_{w_{t}} \end{array}\right]=\left[\begin{array}{ccccc} 0 & I & 0 & 0 & 0\\ 0 & 0 & -R_{t}^{b_{k}}{\left\lfloor {\hat{a}}_{t}-b_{a_{t}}\right\rfloor }_{\times } & -R_{t}^{b_{k}} & 0\\ 0 & 0 & -{\left\lfloor {\hat{\omega }}_{t}-b_{w_{t}}\right\rfloor }_{\times } & 0 & -I\\ 0 & 0 & 0 & 0 & 0\\ 0 & 0 & 0 & 0 & 0 \end{array}\right]\left[\begin{array}{c} \delta \alpha_{t}^{b_{k}}\\ \delta \beta_{t}^{b_{k}}\\ \delta \theta_{t}^{b_{k}}\\ \delta b_{a_{t}}\\ \delta b_{w_{t}} \end{array}\right]+\left[\begin{array}{cccc} 0 & 0 & 0 & 0\\ -R_{t}^{b_{k}} & 0 & 0 & 0\\ 0 & -I & 0 & 0\\ 0 & 0 & I & 0\\ 0 & 0 & 0 & I \end{array}\right]\left[\begin{array}{c} n_{a}\\ n_{w}\\ n_{b_{a}}\\ n_{b_{w}} \end{array}\right]=F_{t}\delta z_{t}^{b_{k}}+G_{t}n_{t}$$
where the 15-dimensional matrix $F_{t}$ is constant over the integration period, such that $F_{d}=\exp\left(F_{t}\delta t\right)$ for a given time-step [12]. With the continuous-time noise covariance matrix:(35)$$Q_{t}=\mathrm{diag}\left(\sigma_{a}^{2},\sigma_{w}^{2},\sigma_{b_{a}}^{2},\sigma_{b_{w}}^{2}\right)\;Q_{d}=\int_{0}^{\delta t}F_{d}(\tau )G_{t}Q_{t}G_{t}^{T}F_{d}(\tau {)}^{T}=\delta tF_{d}G_{t}Q_{t}G_{t}^{T}F_{d}^{T}\approx \delta tG_{t}Q_{t}G_{t}^{T}$$

The covariance matrix $P_{b_{k+1}}^{b_{k}}$ propagates from the initial covariance $P_{b_{k}}^{b_{k}}=0$ as follows:(36)$$P_{t+\delta t}^{b_{k}}=\left(I+F_{t}\delta t\right)P_{t}^{b_{k}}{\left(I+F_{t}\delta t\right)}^{T}+\delta tG_{t}Q_{t}G_{t}^{T},t\in \;[k,k+1]$$

Meanwhile, the first-order Jacobian matrix can be also propagated recursively with the initial Jacobian matrix $J_{b_{k}}=I$ as
(37)$$J_{t+\delta t}=\left(I+F_{t}\delta t\right)J_{t},t\in [k,k+1].$$

#### 2.3.2. Backend Optimization

The state variables vector of the whole system can be defined by
(38)$$X=\left[x_{0},x_{1},\ldots x_{n},x_{c}^{b},\lambda_{0},\lambda_{1},\ldots \lambda_{m}\right]$$
(39)$$x_{k}=\left[p_{b_{k}}^{w},v_{b_{k}}^{w},q_{b_{k}}^{w},b_{a},b_{g}\right],k\in [0,n]$$
(40)$$x_{c}^{b}=\left[p_{c}^{b},q_{c}^{b}\right]$$
where $x_{k}$ is the state vector of the IMU at time k, including the position vector $p_{b_{k}}^{w}$, speed vector $v_{b_{k}}^{w}$, body attitude vector in the world coordinate system $q_{b_{k}}^{w}$ and the accelerometer bias vector $b_{a}$ and the gyroscope bias vector $b_{g}$, and the binocular external parameters should be considered when the system is binocular:(41)$$x_{c}^{b}=\left[p_{c_{1}}^{b},q_{c_{1}}^{b},p_{c_{2}}^{b},q_{c_{2}}^{b}\right]$$

n represents the number of key frames within the sliding window, m represents the number of all points within the sliding window and $\lambda_{i}$ as the inverse depth of the first observed feature point. According to the paper [10], the overall residuals can be written as follows:(42)$$\underset{X}{\min}\left\{{∥r_{p}-H_{p}X∥}^{2}+\sum_{k\in B}{∥r_{B}\left({\hat{z}}_{b_{k+1}}^{b_{k}},X\right)∥}_{P_{b_{k+1}}^{b_{k}}}^{2}+\sum_{(l,j)\in C}\rho \left({∥r_{C}\left({\hat{z}}_{l}^{c_{j}},X\right)∥}_{P_{l}^{c_{j}}}^{2}\right)\right\}$$

The residual difference applies the robust kernel function:(43)$$\rho (s)=\left\{\begin{array}{ll} s & s\leq 1\\ 2\sqrt{s}-1 & s>1. \end{array}\right.$$
where the IMU residuals vector are expressed as $r_{B}\left({\hat{z}}_{b_{k+1}}^{b_{k}},X\right)$, which can be calculated through two recursive frames:(44)$$r_{B}\left({\hat{z}}_{b_{k+1}}^{b_{k}},X\right)=\left[\begin{array}{c} \delta \alpha_{b_{k+1}}^{b_{k}}\\ \delta \beta_{b_{k+1}}^{b_{k}}\\ \delta \theta_{b_{k+1}}^{b_{k}}\\ \delta b_{a}\\ \delta b_{g} \end{array}\right]=F_{t}\delta z_{t}^{b_{k}}+G_{t}n_{t}=\left[\begin{array}{c} R_{w}^{b_{k}}\left(p_{b_{k+1}}^{w}-p_{b_{k}}^{w}+\frac{1}{2}g^{w}\Delta t_{k}^{2}-v_{b_{k}}^{w}\Delta t_{k}\right)-{\hat{\alpha }}_{b_{k+1}}^{b_{k}}\\ R_{w}^{b_{k}}\left(v_{b_{k+1}}^{w}+g^{w}\Delta t_{k}-v_{b_{k}}^{w}\right)-{\hat{\beta }}_{b_{k+1}}^{b_{k}}\\ 2{\left[q_{b_{k}}^{w^{-1}}⊗q_{b_{k+1}}^{w}⊗{\left({\hat{\gamma }}_{b_{k+1}}^{b_{k}}\right)}^{-1}\right]}_{xyz}\\ b_{ab_{k+1}}-b_{ab_{k}}\\ b_{wb_{k+1}}-b_{wb_{k}} \end{array}\right]$$
where ${\left[\cdot \right]}_{xyz}$ represents the real 3D coordinates of the taken quaternions. $\left[{\hat{\alpha }}_{b_{k+1}}^{b_{k}},\;{\hat{\beta }}_{b_{k+1}}^{b_{k}},\;{\hat{\gamma }}_{b_{k+1}}^{b_{k}}\right]$ is the projected submeasurement of the IMU.

The visual measurement residuals $r_{C}\left({\hat{z}}_{l}^{c_{j}},X\right)$ can be expressed as the amount of tangent plane error of the 3D points of the world coordinates projected to the surface of the unit sphere [12]:(45)$$r_{C}\left({\hat{z}}_{l}^{c_{j}},X\right)={\left[\begin{array}{ll} b_{1} & b_{2} \end{array}\right]}^{T}\cdot \left({\hat{\bar{P}}}_{l}^{c_{j}}-\frac{P_{l}^{c_{j}}}{∥P_{l}^{c_{j}}∥}\right)$$
(46)$$P_{l}^{c_{j}}=R_{b}^{c}\left(\left(R_{w}^{b_{j}}\left(R_{b_{i}}^{w}\left(R_{c}^{b}\frac{1}{\lambda_{l}}\pi_{c}^{-1}\left(\left[\begin{array}{c} {\hat{u}}_{l}^{c_{i}}\\ {\hat{v}}_{l}^{c_{i}} \end{array}\right]\right)+p_{c}^{b}\right)+p_{b_{i}}^{w}\right)-p_{b_{j}}^{w}\right)-p_{c}^{b}\right)$$

The prior residual can be expressed as $r_{p}$, the partial residual can be expressed as the H matrix after the sliding window marginalization and the Jacobian matrix obtained via matrix decomposition.

## 3. Results

To evaluate the performance of the proposed method, we used our method to conduct ablation contrast experiments on the EuRoc dataset [12] and TUM-VI dataset [41], which mainly compared the accuracy of the traditional optical flow method with our strategy. Meanwhile, in order to prove the advanced nature of our algorithm and the robustness to illumination changes, we compare experiments with advanced visual algorithms on the UMA-VI dataset [13].

### 3.1. EuRoc Dataset

The EuRoc dataset is a classic SLAM dataset collected using an unmanned aerial vehicle (UAV) equipped with a binocular camera and IMU sensors, as shown in Figure 7. The data acquisition time is hard-time synchronized. For fair comparison, we kept the number of feature points at the front end for equal level, and selected 512 Superpoints or 1024 in our system. It should be pointed out that, although a lot of characteristic points were extracted, due to the features of dispersion strategy and deep feature matching results, the final input to the back-end of the feature point number is always comparable with other equal conditions; the typical value in the radius of 15 and the number of characteristic points is 150, so our feature-tracking strategy can always ensure that the characteristics of the front point number are comparable.

To compare the tracking performance, we only compare the mono-VIO system for the other VIO system using optical flow as the front-end tracking method, just like OKVIS, MSCKF, ROVIO [42], vins-mono (vins-m). The Root-Mean-Squared Error (RMSE) of Absolute Trajectory Error (APE) is shown in the Table 1. The points for tracking are shown in Figure 1a,b.

We also changed the number of vins-mono feature detection to experiment, to compare the localization effects under different feature points. In Table 1, Vins-m-150, Vins-m-300 and Vins-m-400 represent changes in the gft numbers to 150, 300 and 400. At the same time, the radii for dispersing feature points are selected as 30, 20 and 15, respectively, in order to achieve more input of feature points in the backend and compare different positioning results. In our method, we experiment with different combinations of points between GFT and SP as 200 + 1024, 50 + 512, 0 + 512, 0 + 1024, where the preceding numbers represent the number of GFT features, while the following numbers represent the number of SP features which are limited to the number of requirements by the topk method. Due to the use of tensorRT to accelerate our inference, we can adopt more feature points for overall Lightglue matching while keeping real time inference. Due to the fact that the front-end based on feature points always extracts new feature points independently of the previous feature extraction, more points are needed to ensure that more points can match the previous ones.

We also analyzed the performance gap between the traditional GFT scheme and the SP + LG method for front-end tracking, especially under conditions of blurry images. For fair testing, we extracted 1024 SP feature points and 1024 GFT feature points on the front and back frames, respectively, and then compared the feature point tracking performance based on Lightglue and optical flow. As shown in Figure 8 and Figure 9, after successful matching, we perform RANSAC-based outlier removal, and then record the feature points on successful tracking and mark them with the same color in the previous and subsequent frames. The comparison result of the successfully tracked points under image blur is shown in Table 2.

### 3.2. TUM-VI Dataset

Next, we tested our algorithm on the TUM-VI dataset. TUM-VI is a stereo vision-inertial dataset recorded with a handheld device, and we selected five sequences: corridor4, corridor5, room1, room2 and room5 for testing. We tested our algorithm using mono and stereo image data with the resolution of 512 × 512 and compared it with vins-mono and vins-fusion. Among them, “Vins-m”, “VF-m” and “VF-s” represent the mono and stereo versions of vins-mono and vins-fusion, with a feature point count of 150. Our mono and stereo algorithms, Mix-VIO-m and Mix-VIO-s, used different proportions of GFT and SP feature points for comparison with these algorithms, with a typical feature point dispersion radius set to 10. The result is shown in the Table 3.

### 3.3. UMA-VI Dataset

We conducted tests using the indoor and ill-change sequences from the UMA-VI dataset, as shown in Figure 10, Table 4 and Table 5. In the experiment, we first test the monocular version of Mix-VIO on the indoor sequence, adjusting the number of mixed optical flow points to 75, 50 and 0, respectively, where 0 represents the use of only SP+LG for front-end tracking. Subsequently, we ran the binocular version of Mix-VIO, using 50 and 30 GFT features combined with 1024 SP features.

It is worth mentioning again that since our system trusts the optical flow features of binocular tracking more, using different numbers of GFT features also affects the quantity of SP features, due to our feature dispersion strategy. Therefore, although we extracted the top 1024 features, the number of features sent to the backend is always controlled and does not overflow due to the success of the LG matching and our feature dispersion strategy. The term “failed” denotes tracking loss after system operation.

Then, on the ill-change sequence, we ran our mix-VIO and conducted experimental comparisons with advanced systems such as Airvo [35] and PLslam [43]. Airvo is a binocular system that performs front-end tracking based on Superpoint feature extraction and Superglue feature matching. In addition to DNN-based features, it also incorporates line features and uses Superpoint to encode these line features, making front-end tracking more robust in varying illumination conditions and achieving good experimental results. PLslam [43] is another excellent system that uses point and line features for tracking. Since line features can provide more texture information and offer more stability against lighting changes compared to point features, both systems can operate robustly under varying lighting conditions.

We also compared the front-end tracking performance of the traditional GFT scheme and the SP + LG method on sequences with drastic changes in lighting. We still selected 1024 SP feature points and 1024 GFT feature points, and then performed tracking, as shown in Figure 11 and Figure 12. The traditional approach completely lost tracking under these conditions, which would lead to divergence of the visual system. However, the SP + LG approach based on DNN was still successful, as shown in Table 6.

## 4. Discussion

### 4.1. EuRoc Dataset Result

On the EuRoc dataset, our system achieved comparable localization results to vins-mono, particularly standing out in the mix-VIO (50 + 512) configuration, highlighting the success of our hybrid strategy. We also observed that due to the high-speed mobility of drones, images were somewhat blurred, especially in difficult sequences like MH_04, MH_05, V1_03 and V2_03. When only using SP + LG for front-end tracking, the localization results were not very satisfactory. This is due to the inherent weakness of feature extraction and matching against image blur in feature-based methods, which is worth further investigation (as shown in Table 2). However, incorporating direct methods like optical flow into the front-end can still enhance performance, as evidenced by our system. In sequences where image motion was relatively stable, systems that used only DNN-based feature extraction and matching strategies achieved better results, such as in the V1_02 sequence. It is also evident that more features do not necessarily equate to better outcomes, as shown in the vins-m-150, vins-m-300 and vins-m-400 experiments. Despite increased feature count (and a reduced inhibition radius), the localization results did not improve and even led to tracking failures in the V1_03 dataset (as shown in Table 1). This was due to an over-density of features skewing the optimization problem, and when dense feature optical flow tracking failed, the system had to replenish features, causing observational breaks between adjacent image frames. It prompts further reflection on how an optimal feature-tracking strategy should be designed, considering the overall effect on the image.

### 4.2. TUM-VI Dataset Result

On the TUM-VI dataset, we mainly conducted feature point ablation experiments based on different quantities of feature point mixtures. It can be observed that the accuracy of vins-fusion, both in stereo and mono modes, is lower than the vins-mono. In contrast, our Mix-VIO algorithm, after incorporating DNN-based feature detection and matching methods, shows an overall improvement in accuracy compared to vins-m, vf-m and vf-s. However, due to the relatively stable environment and minimal strong lighting changes in the TUM-VI dataset, the accuracy improvement in DNN-based feature extraction and matching schemes is somewhat limited compared to traditional optical flow-based feature matching pipelines.

### 4.3. UMA-VI Dataset Result

On the UMA-indoor sequence, the monocular version of mix-VIO performed with almost higher precision than vins-mono. Additionally, due to some lighting changes and image blur in sequences such as hall-rev-en and hall23-en, vins-mono’s localization results were divergent, whereas our Mix-VIO-m (50 + 1024) system achieved the best average ATE localization results. When switching to the binocular version, due to the robustness of stereo vision, the vins-stereo and vins-fusion versions achieved better localization results. However, it is important to note that on the hall23-en sequence, vins-stereo experienced divergence after running for a while, but our two binocular versions operated robustly.

On the UMA-ill-change sequence, in addition to the same parameter design and experimental comparisons as the indoor sequence, we also compared the binocular version with the Airvo and PLslam systems. Due to the severe lighting changes, which could lead to tracking losses when using only optical flow methods for front-end tracking (as shown in Table 5, all VINS-M-150 failed), we modified some of the VINS code. This allowed the vins-mono system to skip frames with lighting changes or when the lights were turned off in the UMA-VI dataset, and resume tracking in subsequent frames. We call this experimental system vins-m-150 (ours), and it has shown comprehensive advantages over traditional optical flow tracking systems. But it still failed on the conf-csc3 sequence because it was unable to track successfully when restoring illumination again, as shown in our Figure 11. Compared to systems like the Airvo , which uses SP + SG and line feature tracking, our system achieved superior localization results.

### 4.4. Time Consumption

We compared the computational speeds of SP + LG (SuperPoint + LightGlue), SP + SG (SuperPoint + SuperGlue) and GFT + opt-flow (GFT + optical flow) in point detection and matching, including the inference versions accelerated via TensorRT C++ and the original Python versions as shown in Table 7. The SP point number is set to 1024 and the GFT point number is set to 150, as used in our previous experiment. The test was conducted on a computer with an RTX 4060 mobile graphics card, a 13900KF processor and 32 GB of RAM.

As shown in Table 7, the point inference network and feature matching network have achieved approximately three times faster acceleration compared to the original version after TensorRT c++ acceleration. Meanwhile, Lightglue has an inference speed about three times faster than Superglue, which can meet the needs of real-time applications.

## 5. Conclusions

In this paper, we introduce mix-VIO, a monocular and binocular visual-inertial odometry system that integrates a hybrid depth feature extractor Superpoint, a sparse feature matcher Lightglue and traditional handcrafted keypoint detection and optical flow tracking. The system utilizes TensorRT to accelerate deep learning inference, enabling rapid feature extraction and matching. It combines traditional optical flow methods with deep learning-based feature matching to enhance front-end tracking performance under rapid camera motion and varying environmental lighting. In the backend, it employs a sliding window and bundle adjustment (BA) for local map optimization and pose estimation. We conducted extensive testing of our system on the EuRoc, TUM-VI and UMA-VI datasets, demonstrating outstanding results in both accuracy and robustness. Future work will explore tracking strategies under image blur and incorporate loop closure detection to further enhance the system’s robustness and achieve a superior SLAM solution.

## Figures and Tables

**Figure 1 sensors-24-05218-f001:**
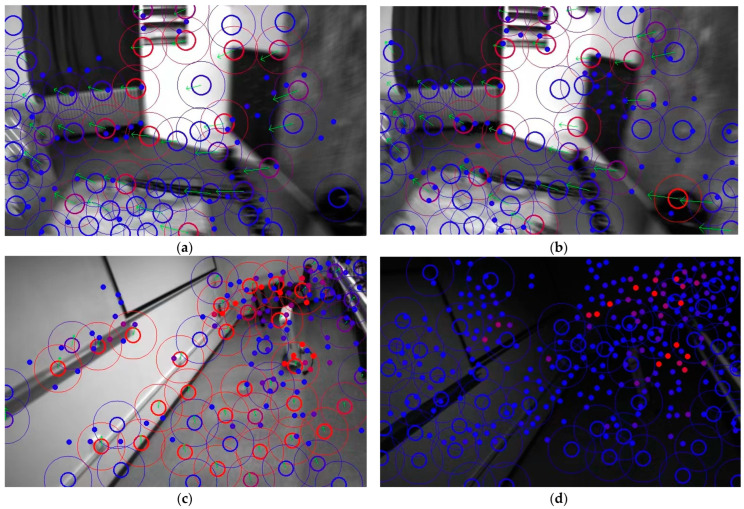
(**a**,**b**) show adjacent images from the EuRoc dataset [12], where noticeable blurring occurs between images. (**c**,**d**) display adjacent images from the UMA-VI dataset [13], highlighting significant changes in lighting between the two images. Blue indicates fewer tracking instances, representing initial feature point, while red indicates frequent successful feature matches due to multiple trackings. Points with rings illustrate results obtained through traditional feature extraction and optical flow matching; the inner circle radius represents the suppression radius for SP features, and the outer circle radius for traditional features, thus dispersing the feature points. The green arrows point to the positions of the points in the previous frame from current frame. Points without rings represent SP features, with 1024 features extracted in the image. It can be observed that in (**a**,**b**), despite the blurring, optical flow matching still matches many feature points, but SP + LG largely fails. In (**c**,**d**), due to drastic illumination changes, optical flow matching fails, but SP + LG still successfully matches many feature points. The traditional approach and deep learning approach complement each other, achieving better tracking results.

**Figure 2 sensors-24-05218-f002:**
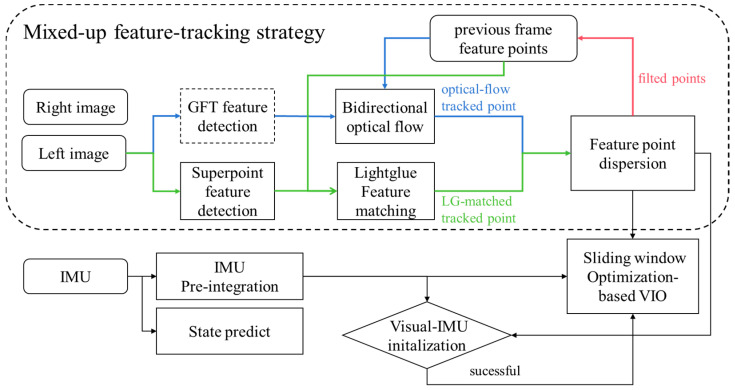
Mix-VIO system overview.

**Figure 3 sensors-24-05218-f003:**
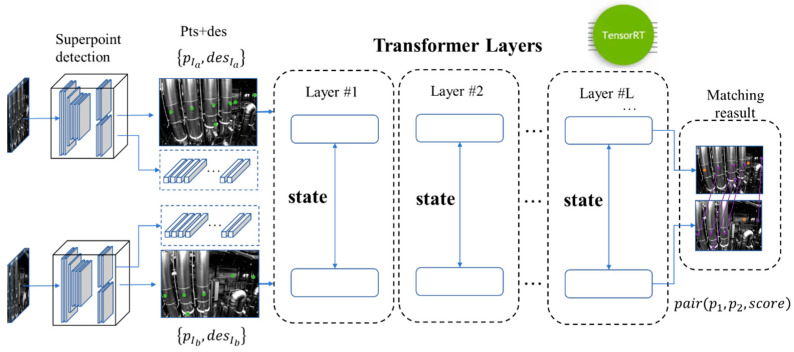
Mixed-up feature-tracking pipeline overview.

**Figure 4 sensors-24-05218-f004:**
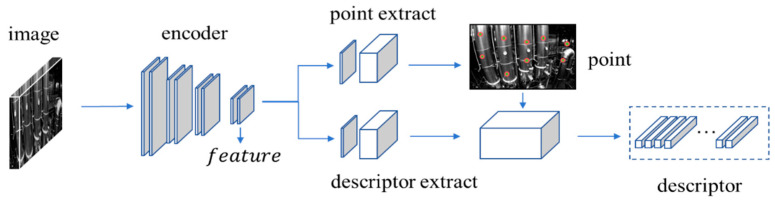
Superpoint network architecture.

**Figure 5 sensors-24-05218-f005:**
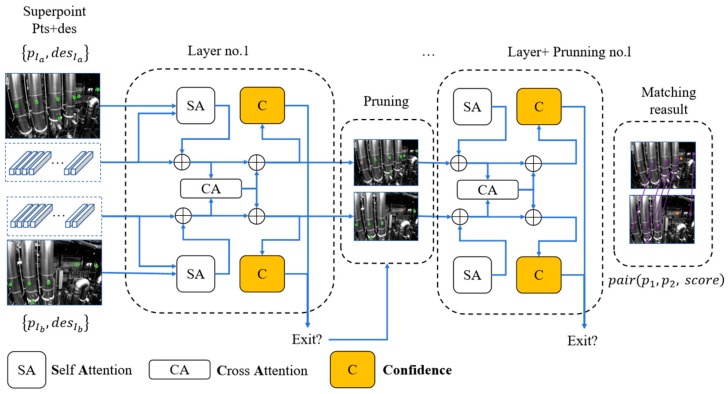
Lightglue network using Superpoint as the input.

**Figure 6 sensors-24-05218-f006:**
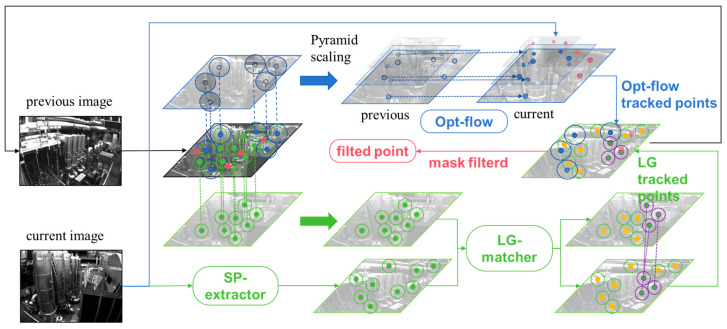
The process of point tracking and the hybrid feature-tracking strategy. The red-colored points represent points where optical flow tracking fails; green points represent Superpoint features which are matched using Lightglue; blue points represent points tracked by optical flow. It is noteworthy that yellow Superpoint points, although unmatched, are added to the system if the number of feature points does not reach the threshold, serving as a basis for feature matching in the next frame.

**Figure 7 sensors-24-05218-f007:**
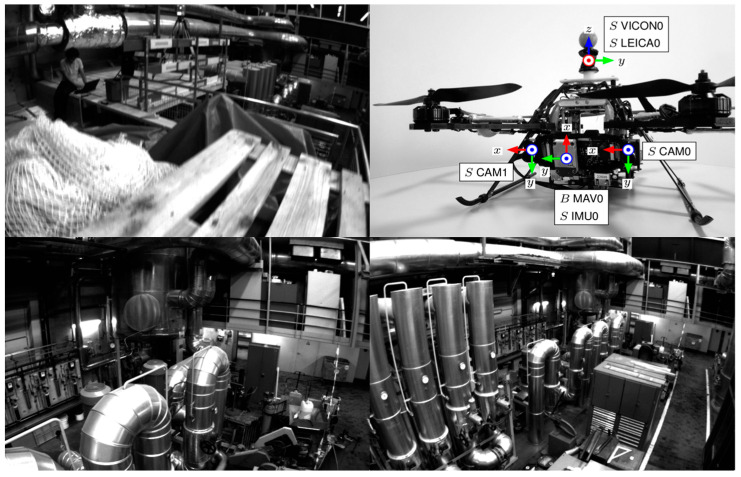
Sequences (Machine House, MH) and collection equipment in EuRoc dataset.

**Figure 8 sensors-24-05218-f008:**
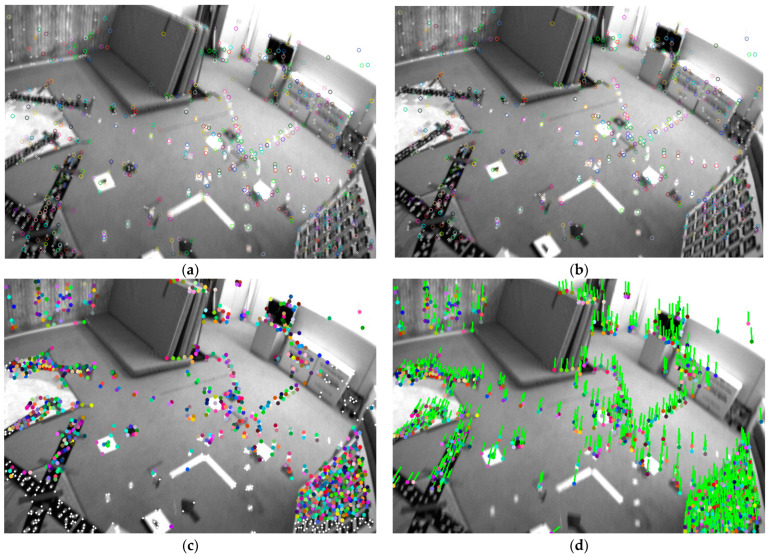
Comparison of SP + LG (**a**,**b**) and GFT+optical flow (**c**,**d**) methods under image blur caused by fast camera movement speed. To distinguish between the two, connect the points matched by the optical flow method with lines and represent the SP points with hollow circles.

**Figure 9 sensors-24-05218-f009:**
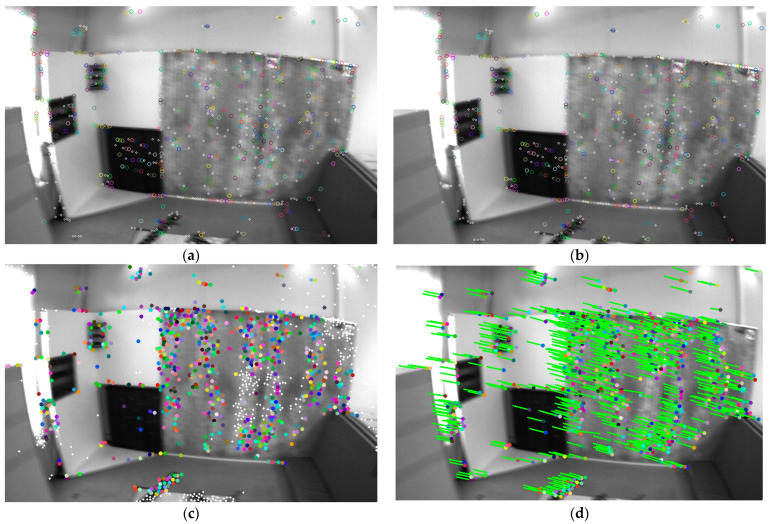
Comparison of another set of SP+LG (**a**,**b**) and GFT + optical flow (**c**,**d**) methods under image blur caused by rapid camera movement.

**Figure 10 sensors-24-05218-f010:**
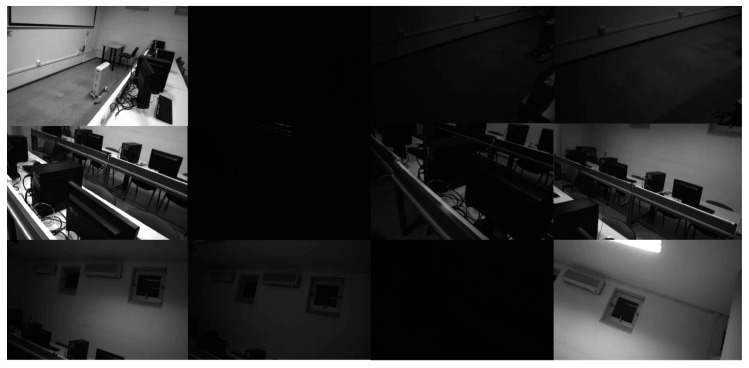
Sequences (ill-change) in UMA-VI dataset. We selected some representative images, and in the actual sequence, several images are kept in low-light conditions. From left to right, the lighting in each row gradually dims and then is turned back on.

**Figure 11 sensors-24-05218-f011:**
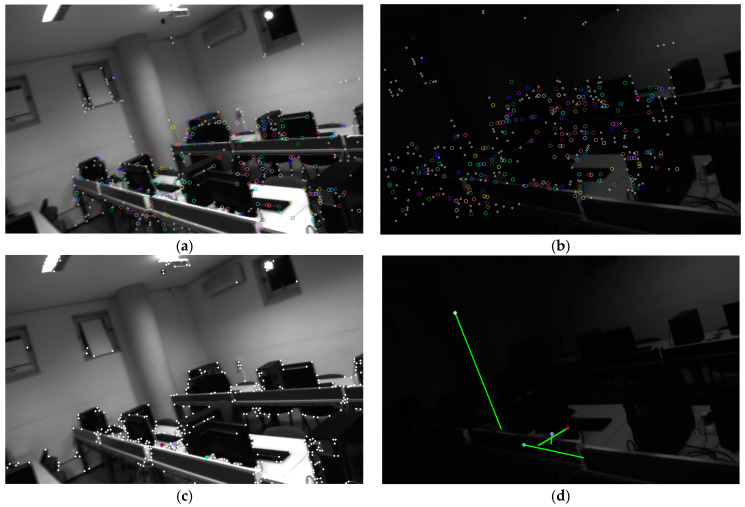
Comparison of another set of SP + LG (**a**,**b**) and GFT + optical flow (**c**,**d**) methods under illumination variations caused by the lighting change. Even if completely dark images are skipped, the optical flow method still cannot track the results of the previous and subsequent frames, which will cause the VIO system to fail.

**Figure 12 sensors-24-05218-f012:**
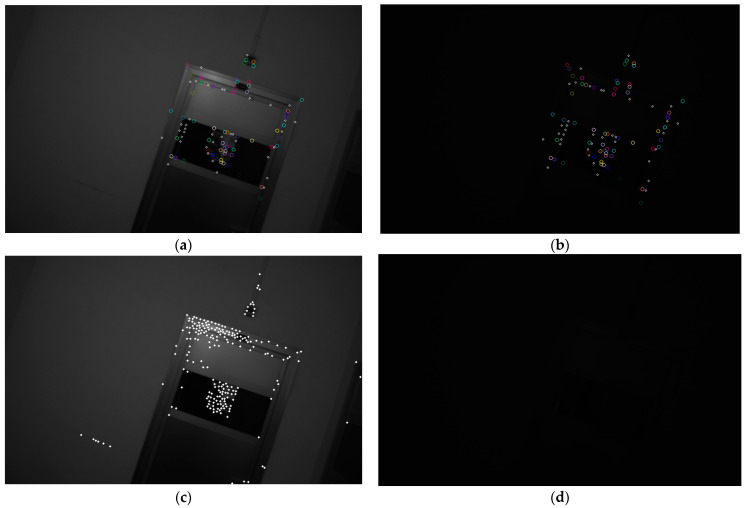
Comparison of another set of SP + LG (**a**,**b**) and GFT + optical flow (**c**,**d**) methods under illumination variation caused by the lighting change in the UMA-VI dataset. Although it is no longer possible to clearly distinguish the contour, the SP + LG-based method can still achieve good tracking results.

**Table 1 sensors-24-05218-t001:** APE RMSE in EuRoc datasets in meters.

EuRoc	MH01	MH02	MH03	MH04	MH05	V101	V102	V103	V201	V202	V203	Average
OKVIS	0.33	0.37	0.25	0.27	0.39	0.094	0.14	0.21	0.090	0.17	0.23	0.231
MSCKF	0.42	0.45	0.23	0.37	0.48	0.34	0.2	0.67	0.1	0.16	0.34	0.341
ROVIO	0.21	0.25	0.25	0.49	0.52	0.10	0.10	0.14	0.12	0.14	0.23	0.231
Vins-m-150	0.15	0.15	0.22	0.32	0.30	0.079	0.11	0.18	0.08	0.16	0.27	0.183
Vins-m-300	0.16	0.13	0.14	0.18	0.33	0.069	0.12	0.16	0.24	0.13	0.16	0.165
Vins-m-400	0.14	0.10	0.08	0.17	0.22	0.066	0.096	failed	0.11	0.11	0.20	-
Mix-VIO (200 + 1024)	0.17	0.12	0.07	0.30	0.25	0.070	0.096	0.13	0.10	0.070	0.13	0.137
Mix-VIO (50 + 512)	0.10	0.13	0.14	0.22	0.35	0.063	0.097	0.15	0.063	0.070	0.12	0.136
Mix-VIO (0 + 512)	0.23	0.18	0.2	0.32	0.36	0.090	0.12	0.15	0.074	0.10	0.49	0.210
Mix-VIO (0 + 1024)	0.22	0.16	0.16	0.27	0.26	0.073	0.11	0.21	0.10	0.10	0.43	0.190

**Table 2 sensors-24-05218-t002:** Comparison of the successfully tracked points under image blur.

	Number	Proportion
SP + LG	415	415/1024 ≈ 41%
GFT + opt-flow	743	743/1024 ≈ 73%

**Table 3 sensors-24-05218-t003:** APE RMSE in TUM-VI datasets in meters.

TUMVI	Corridor4	Corridor5	Room1	Room2	Room5	Average
Vins-m	0.25	0.77	0.07	0.07	0.20	0.272
VF-m	0.26	0.80	0.10	0.07	0.21	0.288
VF-s	0.20	0.88	0.09	0.19	0.14	0.300
Mix-VIO-m (40 + 1024)	0.31	0.80	0.10	0.07	0.18	0.292
Mix-VIO-m (100 + 1024)	0.28	0.67	0.07	0.06	0.15	0.246
Mix-VIO-s (80 + 1024)	0.13	0.66	0.10	0.22	0.29	0.280
Mix-VIO-s (100 + 1024)	0.16	0.55	0.11	0.19	0.31	0.264
Mix-VIO-s (150 + 1024)	0.08	0.68	0.11	0.17	0.22	0.252

**Table 4 sensors-24-05218-t004:** RMSE in UMA indoor datasets in meters.

UMA	Indoor					
	Class-En	Hall1-En	Hall1-Rev-En	Hall23-En	Third-Floor-En	Average
mono						
Vins-m (150)	0.11	0.35	failed	failed	0.36	-
Mix-VIO-m (75 + 1024)	0.20	0.31	0.25	0.29	0.32	0.274
Mix-VIO-m (50 + 1024)	0.11	0.22	0.24	0.23	0.30	0.22
Mix-VIO-m (0 + 1024)	0.31	0.22	0.25	0.29	0.32	0.26
stereo						
Vins-s (50)	0.12	0.14	0.16	drift	0.31	-
Mix-VIO-s (50 + 1024)	0.14	0.26	0.17	0.32	0.26	0.23
Mix-VIO-s (30 + 1024)	0.14	0.18	0.17	0.32	0.30	0.222

**Table 5 sensors-24-05218-t005:** RMSE in UMA illumination-change datasets in meters.

UMA	Illumination-Change				
	Class-Eng	Conf-Csc2	Conf-Csc3	Third-Floor-Csc2	Average
mono					
Vins-m-150	failed	failed	failed	failed	-
Vins-m-150 (ours)	0.11	0.26	failed	0.18	-
Mix-VIO-m (75 + 1024)	0.26	0.26	0.28	0.18	0.245
Mix-VIO-m (50 + 1024)	0.11	0.27	0.28	0.18	0.21
Mix-VIO-m (0 + 1024)	0.31	0.26	0.29	0.17	0.2575
stereo					
Airvo [35]	0.52	0.16	-	0.13	-
PL-slam [43]	2.69	1.59	-	6.06	-
Vins-s (50 ours)	drift	0.23	0.099	0.14	-
Mix-VIO-s (50 + 1024)	0.17	0.16	0.095	0.094	0.15225
Mix-VIO-s (30 + 1024)	0.14	0.20	0.098	0.091	0.1318

**Table 6 sensors-24-05218-t006:** Comparison of the successfully tracked points under illumination variation.

	Number	Proportion
SP + LG	133	133/1024 ≈ 13%
GFT + opt-flow	0	0/1024 ≈ 0%

**Table 7 sensors-24-05218-t007:** Time consumption of methods.

Time	Point-Detection	Point-Matching
SP + LG	9.2 ms	16.9 ms
SP + LG (TRT acc)	2.7 ms	3.5 ms
SP + SG	9.2 ms	42.3 ms
SP + LG (TRT acc)	2.7 ms	12.8 ms
GFT + opt-flow	3.1 ms	7.0 ms

## Data Availability

Data are contained within the article.

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
