# Peer review of "Mix-VIO: A Visual Inertial Odometry Based on a Hybrid Tracking Strategy"

_sensors, 2024, doi:10.3390/s24165218_

Round 1

Reviewer 1 Report

Comments and Suggestions for Authors

In this work, the authors present a Mix-vio, a monocular and binocular visual-inertial odometry, 8 to address the issue where conventional visual front-end tracking often fails under dynamic lighting and image blur conditions. Here some minor issues which I encourage the authors to consider:

Revisions:

1 For the event-based SLAM their references are missing and in my opinion these works are highly related with the proposed approach since the inertial data could be related with the camera events image. In nay case discussion about this can be a nice complement for the current manuscript.

2 The author should consider additional testing in the Kitti or TUM datasets since those are two of the standards for SLAM systems, this is important since those comparison can support the real performance of the proposed approach:

https://www.cvlibs.net/datasets/kitti/

https://cvg.cit.tum.de/data/datasets/rgbd-dataset

Comments on the Quality of English Language

3 There are some minor grammatical/style errors. A grammar/style revision has to be carried out before the manuscript can be considered for publication.

Author Response

We feel great thanks for your professional review work on our article. According to your nice suggestions, we have made extensive corrections to our previous manuscript, the detailed corrections are listed below.

S1. For the event-based SLAM their references are missing and in my opinion these works are highly related with the proposed approach since the inertial data could be related with the camera events image. 
In nay case discussion about this can be a nice complement for the current manuscript.

R1.In the introduction, we added 2 event-based VIO references and 2 muilt-camera references and introduce them in some discussion.

S2 The author should consider additional testing in the Kitti or TUM datasets since those are two of the standards for SLAM systems, 
this is important since those comparison can support the real performance of the proposed approach:

R2.In the absence of strong illumination changes in the image, our system can achieve better results compared to traditional optical flow tracking conditions, but the improvement is not significant

S3. There are some minor grammatical/style errors. A grammar/style revision has to be carried out before the manuscript can be considered for publication.

R3.We have reorganized most of the languages and made some grammar modifications. In the future, we will seek professional institutions to polish and correct language issues

Reviewer 2 Report

Comments and Suggestions for Authors

Mix-vio: A visual inertial odometry based on a hybrid tracking strategy

This paper presents a deep learning-based feature extraction and integrated it with optical flow and tracking to address dynamic lighting and motion blur problems in monocular and binocular visual-inertial odometry (VIO), a branch of simultaneous localization and mapping (SLAM). Local map optimization and pose estimation were based on sliding window and bundle adjustments (BA). For method validation, experiments under dynamic illumination and blur were conducted. The paper is generally well prepared and written. The devised method and the corresponding experiments are technically sound. So, the paper may be considered for publication in Sensors, subject to the following minor concerns are properly addressed.

- Line 94: Please elaborate the key differences between the proposed scheme and [10, 16-19]. The authors may combine this explanation with that in Line 155 to 170 (and 180-186) and dedicate the paragraph to the new sub-section, e.g., Main Contribution. However, those in line 187 to 191 are not the main contribution and can be moved to “Results” section.

- Figure 2: Along with the arrows in the diagram, data type, their dimensions, and notations in the text (those expressed in key equations), etc., should be added for better understanding and clear references to corresponding sub-sections. The diagram can be enlarged to make space for the suggested inclusion. The same applies to Figures 3 – 5. Moreover, the links between each module are not consistent. For example, the o/p of “mixed-up feature-tracking” module is feature point dispersion, while that described in the following subsection is the “matching result.” The above suggestion could help result this and similar issues (e.g., descriptor and a cubic box in Figure 4, how was the IMU in Figure 2 initialized and proceeded as per section 2.3, etc.). Please consider.

- A number of proofs are presented. However, those regarding, i.e., 1) Deep feature detection and matching pipelines can resolve illumination variations and 2) combining optical flow tracking with deep feature extraction make it more robust against image blur are conjectured (Line 310-321) but not properly proved or asserted. Please clarify.

- Those presented in Line 380-398 are not pseudocode. Please follow a more standard format.

- Quite a few notations/ variables in Eq (23-47) miss their descriptions (whether vector or matrix) and their dimensions. This makes comprehension unnecessarily difficult, e.g., F, delta, z, G and n in Eq. (35) and (45), etc. Please resolve this issue throughout. Alternatively, if the original proofs are referred to [12, 13], the authors may consider shortening them and focus to only those vital to the present work.

- Section 3: Please provide experimental details on how the illuminations were systematically varied and the same for image blurring.

- Some references do not contain complete information, e.g., missing authors. Please revise.

- Subscript in some variables in some equations are missing. Please check.

Comments on the Quality of English Language

- Many tortured phrases and unusual terms were detected and made the texts lost in translation. Please let me not give any example as it may risk the authors correcting only the given one. Please resolve the issue throughout the manuscript and highlight (or list) all the corrected words in the revision form.

Author Response

We feel great thanks for your professional review work on our article. According to your nice suggestions, we have made extensive corrections to our previous manuscript, the detailed corrections are listed below.

S1.Line 94: Please elaborate the key differences between the proposed scheme and [10, 16-19]. The authors may combine this explanation with that in Line 155 to 170 (and 180-186) and dedicate the paragraph to the new sub-section,
 e.g., Main Contribution. However, those in line 187 to 191 are not the main contribution and can be moved to “Results” section.

R1.We described our system after [10, 16-19] and added our differences from these systems in the contribution summary of the introduction. 
We modified our contribution and placed the results section in the results section

S2. - Figure 2: Along with the arrows in the diagram, data type, their dimensions, and notations in the text (those expressed in key equations), 
etc., should be added for better understanding and clear references to corresponding sub-sections. 
The diagram can be enlarged to make space for the suggested inclusion. The same applies to Figures 3 – 5. 
Moreover, the links between each module are not consistent. 
For example, the o/p of “mixed-up feature-tracking” module is feature point dispersion, while that described in the following subsection is the “matching result.”
 The above suggestion could help result this and similar issues (e.g., descriptor and a cubic box in Figure 4, how was the IMU in Figure 2 initialized and proceeded as per section 2.3, etc.). Please consider.

R2.Yes, we redraw the figure and change the describtion, enlarge the figure.

S3. A number of proofs are presented. However, those regarding,
 i.e., 1) Deep feature detection and matching pipelines can resolve illumination variations and 
2) combining optical flow tracking with deep feature extraction make it more robust against image blur are conjectured (Line 310-321) but not properly proved or asserted. Please clarify.

R3.We have changed the description of these statements. just "almost" "likely"

S4. Those presented in Line 380-398 are not pseudocode. Please follow a more standard format.

R4.We modified the pseudocode and wrote it in standard mode

5.Quite a few notations/ variables in Eq (23-47) miss their descriptions (whether vector or matrix) and their dimensions. 
This makes comprehension unnecessarily difficult, e.g., F, delta, z, G and n in Eq. (35) and (45), etc. 
Please resolve this issue throughout. Alternatively, if the original proofs are referred to [12, 13], the authors may consider shortening them and focus to only those vital to the present work.

R5.Yes, we resolve the issue throughout. Add the (23-47) descriptions.

S6.Section 3: Please provide experimental details on how the illuminations were systematically varied and the same for image blurring.

R6.Yes, we add experimental details in the 3.Results and the 4.Discussision, such as figure 7-11.

S7.Some references do not contain complete information, e.g., missing authors. Please revise.

R7.We add some references and downloaded from Google Scholar and references exported in Endnote format using MDPI.

S8.Subscript in some variables in some equations are missing. Please check.

R8.Yes, we check the equations and rewrite them. 

S9.Comments on the Quality of English Language

R9. We have reorganized most of the languages and made some grammar modifications. In the future, we will seek professional institutions to polish and correct language issues

Round 2

Reviewer 1 Report

Comments and Suggestions for Authors

Authors have properly addressed the reviewers comments. I don't have further remarks.